# Does Less Pain Predict Better Quality of Life among Malaysian Patients with Mild–Moderate Knee Osteoarthritis?

**Salma Yasmin Mohd Yusuf** [1]**, Mazapuspavina Md-Yasin** [1,*] **and Mohd Fairudz Mohd Miswan** [2]

[1] Department of Primary Care Medicine, Faculty of Medicine, Sungai Buloh Campus,
Universiti Teknologi MARA, Jalan Hospital, Sungai Buloh 47000, Malaysia; salmasoton@gmail.com
[2] Department of Orthopaedic, Faculty of Medicine, Sungai Buloh Campus, Universiti Teknologi MARA,
Jalan Hospital, Sungai Buloh 47000, Malaysia; fairudz@uitm.edu.my
\* Correspondence: puspavina@uitm.edu.my; Tel.: +60-19-356-4020

**Abstract:** This study aims to identify the relationship between knee functional status and Health-Related QoL (HRQoL) in mild to moderate knee osteoarthritis (OA) patients and to ascertain which subdomain of knee functional status best predicts good HRQoL. A cross-sectional study was conducted in an orthopaedic clinic of a tertiary hospital in Malaysia. Patients aged 40–75 years old with mild–moderate primary knee OA were recruited. The Knee Injury and Osteoarthritis Outcome Score (KOOS) and SF-36 questionnaires were used to measure knee functional status and HRQoL, respectively. Subdomains of KOOS include "function in daily living", "function in recreational activities", "pain", "symptom", and "knee-specific quality of life". Subdomains for SF-36 are Physical Component Summary (PCS) and Mental Component Summary (MCS). Overall, 290 patients fulfilled the inclusion criteria of the study, with a mean age of 66.8 years old ($\pm$7.06). Majority were female (57.6%) and Malay (79.7%). The relationships between all KOOS and HRQoL subdomains were significant. "Pain" contributed most towards better physical HRQoL ((PCS) Adj. B (95% CI); 0.063 (0.044, 0.169)), while "function in daily living" contributed most towards better mental HRQoL ((MCS) Adj. B (95% CI); 0.624 (0.478, 0.769)). Thus, better HRQoL was related to better pain control and improved "function in daily living" in these patients.

**Keywords:** knee osteoarthritis; functional status; health-related quality of life; KOOS; SF-36

## 1. Introduction

The Global Burden of Disease 2010 Study reported musculoskeletal disorders as the second greatest cause of disability, in which osteoarthritis (OA) is one of the causes. Among various types of OA, 83% is represented by knee OA [1]. Currently, the prevalence of knee OA in Malaysia is estimated to be 10–20% of the total adult population [2,3]. As knee OA progresses, pain and disability lead to decrements in knee function [4]. The reduced functional status causes disability, leading to impairment in individuals' function, whether in daily living, economic-sustaining activities, or recreation [5–7]. Disability from knee pain poses substantial economic impacts and is a major health problem globally [8,9]. Thus, knee functional status has become an important measure in understanding the effects of knee OA on physical impairment and disability [4]. With increasing survival age of the world population and increasing obesity, knee OA is the major reason for knee replacement.

Health-Related Quality of Life (HRQoL) is how a health state impacts on an individual's ability to function, and one's perception of well-being in the physical, mental, and social domains of life [10–12]. In knee OA, physical restrictions from the pain and disability directly influence other aspects of life, for example, their social interactions, mental health, and quality of sleep, which in turn impact their HRQoL [13–15]. Improvement in knee OA HRQoL is expected to lessen its economic burden, encourage socialisation in physical recreational activities, and circumvent mental health issues related to knee OA [16–18].

Worldwide, there are limited studies that identify the relationship between functional status and HRQoL in patients with mild to moderate knee OA. Majority of the studies were performed on severe knee OA patients planned for surgery. For example, studies investigated symptomatic knee OA after ACL reconstruction [19], and functional status improvements with HRQoL after total knee arthroplasty [20,21]. In addition, existing studies of various severity only looked at certain target populations [22–24], at selected subdomains of functional status with or without HRQoL, and studies only on HRQoL without measuring the functional status. Furthermore, these studies only measured "pain" and "knee-specific quality of life" at different sides of the knee, either unilateral and/or bilateral [25], HRQoL comparing knee pain with or without knee OA [26], and looking at all subdomains of functional status without HRQoL [27].

To date, only one study was found in Malaysia that investigated functional status in all severities of knee OA. However, it only measured three subdomains of functional status, which were "pain", "function in daily living", and "function in recreational activities", of the studied population [3]. The other functional status subdomains that are under-studied in Malaysia include "symptoms" and "knee-specific quality of life". In addition, HRQoL in knee OA patients, that was performed by Zakaria et al. at two primary healthcare clinics in Klang Valley, did not investigate the relationship between functional status and HRQoL [28].

Therefore, the objectives of this study were: (i) to determine the relationship between all the subdomains of functional status and HRQoL in mild to moderate knee OA in the Malaysian population, and (ii) to ascertain which subdomain of functional status best predicts good HRQoL in this group of patients.

## 2. Materials and Methods

### 2.1. Study Design and Study Population

We conducted a cross-sectional study among patients with mild to moderate knee OA attending an orthopaedic clinic in Selangor, Malaysia. The conduct of the study is outlined in the flowchart presented in Figure 1.

Patients were selected according to the inclusion and exclusion criteria. The inclusion criteria were: (a) aged between 40 and 75 years old, and (b) diagnosed to have mild and moderate knee OA based on the Kellgren–Lawrence (K–L) radiological classification, in which K–L 1–2: Mild, and K–L 3: Moderate [29]. The exclusion criteria were: (a) illiterate and does not understand Malay language, (b) any other forms of lower-limb abnormality such as secondary OA (due to rheumatic or metabolic bone disease) or traumatic injuries, (c) any previous corrective surgery of the lower limbs, and (d) knee OA with severe disability (defined as handicap or needing full assistance in all activities of daily living and social roles).

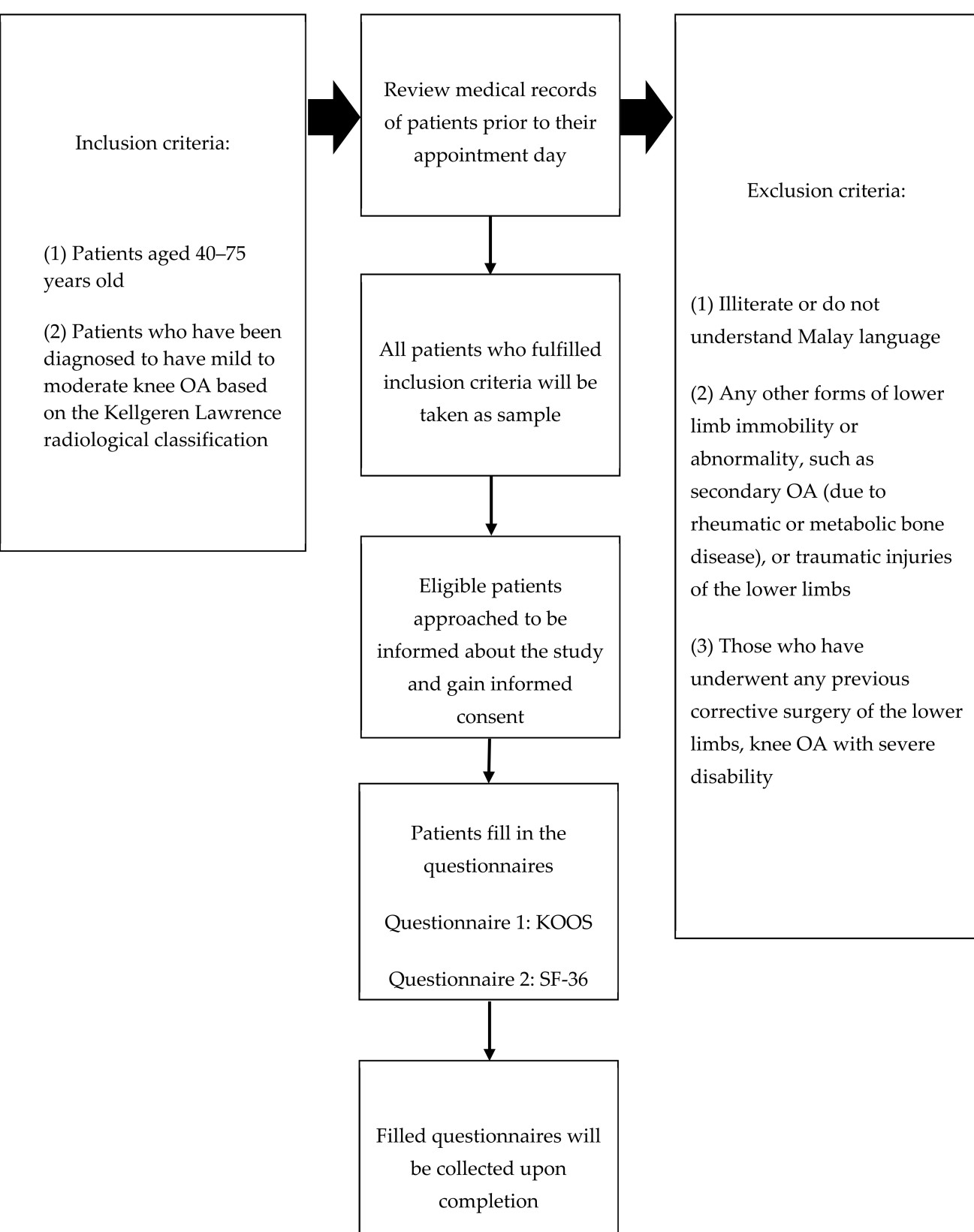

**Figure 1.** Study flow chart.

## 2.2. Study Tools

*Functional status using KOOS*

The validated Malay version of the Knee Injury and Osteoarthritis Outcome Score (KOOS) questionnaire was used to measure functional status [30]. KOOS consists of five subdomains, which are "pain", "other symptoms", such as swelling and restricted range of motion, "function in daily living", "function in recreational activities", and "knee-specific effect on life" [30]. The Cronbach's alpha value ranged from 0.776 to 0.946, while the composite reliability values of each construct ranged between 0.819 and 0.921 [30]. The five subdomains are scored separately using a Likert scale. Each of the five subdomains was calculated as the sum of the items included. Then, the scores were interpreted based on the transformed scale of 1–100, where the higher the score, the better the function, as per the Formulas (1)–(5) [31]. Scoring calculation formulae for KOOS subdomains:

$$\text{PAIN } 100 - \frac{\textit{Mean Score } (P1 - P4) \times 100}{4} = \textit{KOOS Pain} \tag{1}$$

$$\text{SYMPTOMS } 100 - \frac{\textit{Mean Score } (S1 - S5) \times 100}{4} = \textit{KOOS Symptoms} \tag{2}$$

$$\text{ADL } 100 - \frac{\textit{Mean Score } (A1 - A10) \times 100}{4} = \textit{KOOS ADL} \tag{3}$$

$$\text{SPORT/REC } 100 - \frac{\textit{Mean Score } (SP1 - SP5) \times 100}{4} = \textit{KOOS Sport/Rec} \tag{4}$$

$$\text{QOL } 100 - \frac{\textit{Mean Score } (Q1 - Q2) \times 100}{4} = \textit{KOOS QOL} \tag{5}$$

*Health-related quality of life (HRQoL) using SF-36*

The Malay version of SF-36 was chosen to measure the HRQoL as it is the most well-recognised, user-friendly questionnaire for measuring HRQoL [31]. Permission to use the translated Malay version of SF-36 was obtained from the Quality Metric Inc., with license number QM048259. Internal consistency of the Malay version has a Cronbach's $\alpha$ for all of the items of more than 0.07, except for social functioning, in a multi-centre asthmatic and population-based study [32]. This questionnaire consists of 8 health subscales and a total of 36 items [31]. The eight subscales of HRQoL are divided into two subdomains, namely the Physical Component Summary (PCS) and the Mental Component Summary (MCS), with four subscales each, as shown in Table 1. The total score for each subdomain was calculated using the software provided by QualityMetric (QualityMetric's Health Outcomes™ Scoring Software 5.0, Johnston, RI, USA). The scores were interpreted based on the transformed scale of 1–100, where the higher the score, the better the quality of life in that subscale [31].

**Table 1.** Operational definitions used in the study.

| Questionnaire | Variables | Description |
|---|---|---|
| KOOS [26] | Symptom | Questions on symptoms such as swelling, restricted range of motion, and mechanical symptoms. |
| | Pain | Level of pain experienced while bending the knee, straightening the knee, walking on flat surface, standing upright, and at night while in bed. |
| | Function in daily activities | Limitation experienced during function in daily activities. |
| | Function in recreational activities | Limitation experienced during function in recreational activities. |
| | Knee-specific quality of life | Questions on patients' knee-specific quality of life. |

**Table 1.** *Cont.*

| Questionnaire | Variables | Description |
|---|---|---|
| SF-36-PCS [31] | Physical Functioning | Severe and minor physical limitations in extremes of physical activities, including lifting and carrying groceries, climbing stairs, bending, kneeling, stooping, and walking moderate distances. |
| | Role Physical | Physical health-related role limitations, including: (a) limitations in the kind of work or other usual activities, (b) reductions in the amount of time spent on work or other usual activities, (c) difficulty performing work or other usual activities, (d) accomplishing less. |
| | Bodily pain | Comprises 2 items: (a) pertaining to the intensity of bodily pain, (b) measuring the extent of interference with normal work activities due to pain. |
| | General Health | A rating of health (excellent to poor) and four items addressing the views and expectations of his or her health. |
| SF-36-MCS [30] | Vitality | Measure of energy level and fatigue, developed to capture differences in subjective well-being |
| | Social Functioning | Health-related effects on quantity and quality of social activities, asking specifically about the impact of either physical or emotional problems on social activities. |
| | Role emotional | Mental health-related role limitations in terms of: (a) time spent on work or other usual activities, (b) amount of work or activities accomplished, (c) the care with which work or other activities were performed |
| | Mental Health | One or more items from each of four major mental health dimensions (anxiety, depression, loss of behavioural/emotional control, and psychological well-being). |

PCS: Physical Component Summary, MCS: Mental Component Summary.

*2.3. Sample Size Determination*

A study by Zakaria et al. found that the mean (±SD) of HRQoL amongst knee OA patients is 51.88 (±24.11) [28]. The minimum sample required in our study is 89 using the single mean formula below:

$$n = \frac{(Z_{\alpha/2} \; X \; \sigma)2}{\Delta 2} \tag{6}$$

$Z_{\alpha/2} = 1.96$ (critical value confidence interval)
$\sigma = 24.11$ (population standard devation)
$\Delta = 5.0$ (the estimated difference from the population mean)

After considering a 20% attrition rate, our study aimed to approach approximately 108 patients to recruit at least 89 participants.

*2.4. Patient Recruitment, Sampling Method, and Data Collection*

Patients were recruited over 5 months in an orthopaedic clinic from August to December 2018. Researchers attended the clinic the day before clinic day to assess the patients' record based on the appointment book. Data obtained from the electronic system are the patients' demographic details, classification of OA, as well as assessing X-rays of the patients. On clinic days, patients who fulfilled the inclusion criteria were approached in the nurse's assessment room and invited to participate. Participants were given approximately 20 min to complete the self-administered questionnaire. Once the questionnaires were completed, participants were requested to return them directly to the researcher and the questionnaires were checked for completeness.

*2.5. Operational Definitions*

The operational definitions used in the study are summarised in Table 1.

*2.6. Data Entry and Statistical Analysis*

All the data collected were entered and coded into Statistical Package for Social Sciences (SPSS) version 24. Categorical variables were described in numbers and percentages, whereas continuous variables were expressed as mean ± SD for normally distributed data or median with interquartile range (IQR) for non-normally distributed data. To determine the relationship between functional status and HRQoL, simple and multiple linear regression were used.

## 3. Results

Of the 307 knee OA patients who were approached and invited into the study after fulfilling the inclusion and exclusion criteria, 10 (3%) refused to be included in the study, giving reasons such as they were "not interested" or they "did not have the time". A further seven participants were excluded from the final analysis due to incomplete responses. The total participants included in the final analysis was 290, which accounted for a 94.4% response rate.

*3.1. Sociodemographic and Knee OA Profiles of Participants*

Table 2 shows the sociodemographic and knee OA profiles of the participants. The mean age of participants was 66.8 years old (±7.06). Most of the participants were female (57.6%), Malays (79.7%), had a secondary level education (46.2%), were unemployed/retiree (60.3%), had low-income status (75.5%), and were non-smokers (90.7%). Patients with mild knee OA constitute more than half of the study participants (52.1%). There was no family history of knee OA in 74.1% of participants. In terms of treatment received, 51.7% were on topical treatment, 65.9% had not had physiotherapy, and 92.1% were not using a walking stick for walking assistance.

**Table 2.** Sociodemographic characteristics and knee OA profiles of the participants.

| Variables | *n* (%) | Mean (±SD) |
|---|---|---|
| Age (years) | | 66.8 (±7.06) |
| Gender: | | |
| Male | 123 (42.4) | |
| Female | 167 (57.6) | |
| Ethnicity: | | |
| Malay | 231 (79.7) | |
| Chinese | 35 (12.1) | |
| Indian | 24 (8.3) | |
| Education: | | |
| No formal education | 17 (5.9) | |
| Primary level | 74 (25.1) | |
| Secondary level | 134 (46.2) | |
| Tertiary level | 65 (22.4) | |
| Occupational status: | | |
| Desk work | 70 (24.1) | |
| Labour worker | 44 (15.2) | |
| Unemployed | 105 (36.2) | |
| Retiree | 71 (24.1) | |
| Personal income (RM): | | |
| B40 (<4360) | 219 (75.5) | |
| M40 (4360–9619) | 71 (24.5) | |

**Table 2.** *Cont.*

| Variables | *n* (%) | Mean (±SD) |
|---|---|---|
| Smoking status: | | |
| Yes | 27 (9.3) | |
| No | 263 (90.7) | |
| OA Severity: | | |
| Mild | 151 (52.1) | |
| Moderate | 139 (47.9) | |
| Family history of knee OA: | | |
| Yes | 75 (25.9) | |
| No | 215 (74.1) | |
| Current knee OA treatment: | | |
| Not on any medication | 59 (20.3) | |
| Topical treatment | 150 (51.7) | |
| Oral medication | 62 (21.4) | |
| Intra-articular injection | 19 (6.6) | |
| Ever have physiotherapy session: | | |
| Yes | 99 (34.1) | |
| No | 191 (65.9) | |
| Walking assistance: | | |
| No | 267 (92.1) | |
| Walking stick/walking frame | 18 (6.2) | |
| Others/Wheelchair | 5 (1.7) | |

N = 290.

### 3.2. Functional Status Using KOOS among Study Participants

Table 3 shows the overall summary of each subdomain of the KOOS score. The highest score was in "function in daily activities", with a mean score of 53.51 (±13.65), while the lowest score was 41.03 (±18.69) in "knee-specific quality of life".

**Table 3.** Overall summary of KOOS subdomain score.

| No. | Domains | Mean | ±SD |
|---|---|---|---|
| 1 | Symptom | 50.28 | 14.81 |
| 2 | Pain | 49.68 | 14.71 |
| 3 | Function in daily activities | 53.51 | 13.65 |
| 4 | Function in recreational activities | 43.53 | 20.28 |
| 5 | Knee-specific quality of life | 41.03 | 18.69 |

### 3.3. Health-Related Quality of Life of the Participants Using SF-36

The SF-36 quality of life subdomain and subscale scores are shown in Table 4. The highest mean score was in "bodily pain", with 44.6 (±6.7), which means participants felt that the better their bodily pain is, the better their HRQoL. On the other hand, the lowest mean score was in "role-emotional", with 29.8 (±8.8), which means the emotional role was least affected in mild to moderate knee OA participants. It also shows that overall, the Physical Component Summary score was higher than the Mental Component Summary score, 45.2 (±5.3) and 31.4 (±8.7), respectively.

**Table 4.** Overall summary of HRQoL of the patients according to the subdomains of Physical Component Summary (PCS) and Mental Component Summary (MCS) and their subscales.

|  | Mean | SD | Minimum | Maximum |
|---|---|---|---|---|
| Physical Functioning | 41.5 | 8.0 | 21.18 | 57.54 |
| Role-Physical | 36.1 | 7.0 | 21.23 | 57.16 |
| Bodily Pain | 44.6 | 6.67 | 25.71 | 62.00 |
| General Health | 39.9 | 4.2 | 28.46 | 54.61 |
| Physical Component Summary | 45.2 | 5.3 | 32.38 | 62.44 |
| Vitality | 42.8 | 6.1 | 22.89 | 58.54 |
| Social Functioning | 37.6 | 3.3 | 22.25 | 47.31 |
| Role-Emotional | 29.8 | 8.8 | 14.39 | 56.17 |
| Mental Health | 31.2 | 8.6 | 11.63 | 53.48 |
| Mental Component Summary | 31.4 | 8.7 | 8.49 | 48.78 |

*3.4. Multiple Linear Regression*

All the independent variables with a *p*-value less than 0.05 in simple linear regression (SLR) were included in the multiple linear regression (MLR) to adjust for confounding factors. The stepwise method was chosen in the analysis. Any significant variables in the MLR were checked for interaction and multicollinearity. The model was then tested for homoscedascity to test for model fit. The model was also checked for linearity by plotting a scatter plot between residuals and predicted values.

*3.5. Relationship between KOOS and SF-36: Physical Component Summary (PCS) and Mental Component Summary (MCS)*

Table 5 shows the multiple linear regression to determine the relationship between KOOS variables with $p < 0.05$ and the Physical Component Summary (PCS). Data in Table 5 showed that 21.1% of the predictors of PCS were explained by this model. The best predictor was the "pain" score, which contributed 34.5%. The better the improvement of pain, the better the PCS. Pain treatment is the most important treatment in patients with mild to moderate knee OA in improving the physical HRQoL. Table 5 also shows the MLR data to determine the relationship between KOOS variables with $p < 0.05$ and the Mental Component Summary (MCS). Data in Table 5 showed that 21.3% of the predictors of MCS were explained by this model. The best predictor was "function in daily living", which contributed 67.7%. The better the improvement in function in daily living, the better the MCS. The aim of treatment is to improve the functions in daily living, such as getting in/out of the car, going shopping, and performing domestic duties.

**Table 5.** Relationship between functional status KOOS and subdomains of HRQoL: Physical Component Summary (PCS) and Mental Component Summary (MCS).

| Subdomains of HRQoL | Adj. B (95%CI) | Standardised Coefficient Beta | *t* | *p*-Value | R2 |
|---|---|---|---|---|---|
| **\* Physical Component Summary** | | | | | |
| Constant | 2.139 | | 36.494 | | |
| Symptom | 0.029 (0.049, 0.107) | 0.186 | 2.174 | 0.002 | 0.211 |
| Pain | 0.063 (0.044, 0.169) | 0.345 | 3.143 | 0.001 | |
| Function in daily living | 0.168 (0.073, 0.284) | 0.335 | 3.133 | 0.002 | |
| **\*\* Mental Component Summary** | | | | | |
| Constant | 2.071 | | 22.076 | | |
| Function in daily living | 0.624 (0.478, 0.769) | 0.677 | 8.445 | <0.001 | 0.213 |
| Function in recreational activities | 0.237 (0.335, 0.139) | 0.382 | 4.762 | <0.001 | |

Coefficient table MLR (Stepwise method). * Model: R-square = 0.211, Durbin–Watson = 1.526, no problems in multicollinearity, no interaction, model fit. ** Model: R-square = 0.213, Durbin–Watson = 1.333, no problems in multicollinearity, no interaction, model fit.

In comparison, by using the standardised coefficient beta, mental HRQoL had a better impact by improvement of its predictors than the improvement of physical HRQoL by its predictors. In other words, the best predictors from each subdomain showed that MCS will increase by 0.677 units if the "function in daily living" improved by 1 unit and PCS will increase by 0.345 units if the "pain" improved by 1 unit.

## 4. Discussion

To the best of our knowledge, the relationship between functional status and HRQoL in mild to moderate knee OA patients has not been reported. Previous studies were limited to either functional status or HRQoL only, or studies carried out on different target populations. Under functional status, which consists of "symptoms", "pain", "function in daily living", "function in recreational activities", and "knee-specific quality of life" in this study, the highest mean score was the subdomain "function in daily activities". This is comparable to the findings by Foo et al., in which the participants' mean for "function in daily activities" subdomain was approximately similar [3]. The lowest score in functional status in this study was in the subdomain "knee-specific quality of life". This population perceived that their knee health was most affected, however they were still able to perform their functions in daily living. This implies that function in daily living not only involves the knee but needs the functioning of the whole-body system. In addition to the availability of lifestyle adaptation and health facilities in this setting, the ability to use the whole-body system, and not just the knee, during functions in daily living may explain why in this population of mild–moderate knee OA, they have a higher score in this subdomain.

Items related to the PCS showed relatively higher scores compared to the MCS. Among the four subscales comprising the physical component summary, bodily pain (BP) had the highest mean score, while the lowest subscale was "role-physical". As bodily pain is scored at low scores, it shows high levels of pain which impact normal activities, while high scores show lesser pain and lesser impact on normal activities [31]. This means that bodily pain contributed the most to the physical component summary subdomain. This means that the higher the score, the less pain felt by participants. The bodily pain may not be just the knee pain, as this item asks about other physical pains contributing to their response for this item. In the study by Zakaria et al., the highest mean score for the PCS was contributed by "role-physical", with 67.54 (±46.16) [28], and the lowest mean score was "physical functioning", with 51.88 (±24.11). This may be explained by the fact that Zakaria et al.'s study was carried out in the suburban area of Hulu Langat, where the population may be more used to labour work in which mild–moderate knee OA symptoms do not affect them [28].

Amongst the four subscales which make up the MCS, the subscale "vitality" had the highest score. The lowest mean score was in the "role-emotional" (RE) subscale. In the study by Zakaria et al., the highest mean score within the MCS was in "social functioning", with 93.62 (±56.06), while the lowest was in "vitality", with 77.84 (±16.33) [28]. However, the participants scored higher in the MCS as compared to the PCS, in contrast to the findings of this study. Thus, this explains why the mental component subdomain scores in [28] were overall higher than the mental component subdomain scores of this study.

In this study, the mental component score was lower than the physical component score. Other study findings of knee OA patients in which the mental subdomain of HRQoL scored lower than the physical subdomain were, for example, a study in Saudi elders [33] and the study in [34]. This proves that knee OA has a negative bearing on the mental health components of patients.

This study found that there is a significant positive relationship between the subdomains of functional status and the PCS subdomain of HRQoL, as well as the MCS subdomain of HRQoL. It was also found that the factors that best predict each component of the HRQoL include "pain", which best predicted better physical HRQoL (PCS), while the subdomain "function in daily living" best predicted better mental HRQoL (MCS). For PCS, this is shown from the MLR table (Table 5). Subdomains of "symptom", "pain", and

"function in daily living" contributed 21.1% of the variance, of which most was contributed by better pain control, with a standardised beta coefficient of 0.345. Previous studies have proven that worse pain in knee OA causes worse physical HRQoL [22,25–27,35–38]. Therefore, improving pain control will improve the physical HRQoL in these patients.

For MCS, the subdomains of "function in daily living" and "function in recreational activities" contributed 21.3% of the variance, of which "function in daily living" contributed the most (67.7%) towards this subdomain. This finding is in line with the findings by Alves and Bassitt, where it was found that patients' sense of autonomy in performing their daily tasks and social participation have been reported to be significant in translating into elderly patients' quality of life [23]. Limited daily functioning in knee OA patients has been shown to be affected by pain from knee OA [39]. Apart from pain catastrophising, knee confidence and fear of movement affect patients' ability to perform their daily tasks [40]. Additionally, pain and limitations lead to stress, anxiety, and depression in patients [41,42]. HRQoL mental health components have been shown to be affected more than the physical component in selected existing studies on knee OA [33,34]. Thus, improvement of "function in daily activities" is a factor that will improve the mental health components of the HRQoL.

*Strengths, Limitations, and Implications for Clinical Practice and Future Research*

The findings from this study have provided quantitative evidence and added to the body of the knowledge on the relationship between functional status and HRQoL of mild–moderate knee OA patients. A limitation of this study is the setting, which was within an urban tertiary hospital. Furthermore, the convenience sampling method used due to limited time and unavailability of a patient database poses a major limitation, especially in vulnerability towards sampling bias. Additionally, the cross-sectional method used only provided an analysis of the present situation.

For better pain assessments, the findings of this study suggest an improved approach to treating pain that will consequently improve patients' physical subdomain of HRQoL. Concerted efforts by healthcare professionals in addressing the impact on patients' daily functioning will contribute towards improvement of their mental health subdomain of HRQoL. This research used a cross-sectional approach. Therefore, in the future, a longitudinal approach could be applied to gain possibilities of other outcomes. Additionally, further research looking at different geographical settings, such as rural and suburban areas, may be able to provide a greater understanding of HRQoL among patients with mild–moderate knee OA in different settings.

## 5. Conclusions

In conclusion, improvement of overall HRQoL (PCS and MCS) can be achieved by improvement of "symptoms", "pain", "function in daily living", and "function in recreational activities". Mental HRQoL, particularly "function in daily living", had a higher impact on functional status improvement as compared to physical HRQoL. Between all these predictors, improvement of "function in daily living" best predicted the overall HRQoL. It is therefore pertinent that in managing mild to moderate knee OA patients, effort is focused on a holistic approach, emphasizing the mental health of patients as well as addressing their physical health.

**Author Contributions:** Conceptualisation, S.Y.M.Y., M.M.-Y. and M.F.M.M.; methodology, S.Y.M.Y., M.M.-Y. and M.F.M.M.; analysis, S.Y.M.Y.; writing—original draft preparation, S.Y.M.Y. and M.M.-Y.; writing—review and editing, S.Y.M.Y. and M.M.-Y. All authors have read and agreed to the published version of the manuscript.

**Funding:** This research received no external funding.

**Institutional Review Board Statement:** Ethical approval was granted for this study by the National Medical Study Medical Ethics and Study Committee (MREC) (NMMR-17-1578-36441), and the University Study Ethics Committee, ID: 600-IRMI (5/1/6).

**Informed Consent Statement:** Informed consent was obtained from all subjects involved in the study.

**Data Availability Statement:** Not applicable.

**Conflicts of Interest:** The authors declare no conflict of interest.

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
