# Peer review of "Does Less Pain Predict Better Quality of Life among Malaysian Patients with Mild–Moderate Knee Osteoarthritis?"

_clinpract, doi:10.3390/clinpract12020026_

Round 1

Reviewer 1 Report

Thank you for the possibility of reviving the manuscript entitled. “Does less pain predict the better quality of life in mild-moderate knee osteoarthritis patients?”. Overall, data is presented understandably. However, the main text needs extensive editing in style and English. More detailed comments are below:

- please rewrite the introduction and discussion, avoiding sentences with the same context and the exact words all over again ( ex—disability (lines: 40 and twice in line 42; or Zakaria et al. (lines 348, 351, 354).

- please avoid unssesecary spaces and separations between sentences, but add them where needed ex. Line 348 “….8.80).In the study….”                              

- please explain the HRQoL abbreviation in the abstract

- please shorten the Abstract up to 200 words (please see the journal guidelines) and make it a single paragraph. Moreover, please avoid too many numbers in the abstract and instead focus on words.

-please avoid subparagraphs/ headliners  in the discussion part

- In Table 2 in Variable Current knee Oa treatment total number of patients is 291 instead of 290. Please double-check the values.

- In table 4 please change the “Std deviation” for SD. Is it possible to add the intervals +/-?. What is the goal of showing 25-50-75-percentile if you are not commenting/discussing it?

- please edit the reference style according to the journal guidelines

-please specify reference 4, and if its an online source, the date of access

- please edit the citation style in line 94

- please change & for and in line 384

- please add references 43,44 and 45

Author Response

Dear Reviewer 1. Thank you for your comments. Attached are the responses. Thank you. 

Response to Reviewer 1-

No.

Comment

Response

1.       

Thank you for the possibility of reviving the manuscript entitled. “Does less pain predict the better quality of life in mild-moderate knee osteoarthritis patients?”. Overall, data is presented understandably. However, the main text needs extensive editing in style and English.

Thank you for the comments.

We have sent out for proof reading for the manuscript writing.

2.       

please rewrite the introduction and discussion, avoiding sentences with the same context and the exact words all over again ( ex—disability (lines: 40 and twice in line 42; or Zakaria et al. (lines 348, 351, 354).

The introduction has been re-written as suggested and highlighted from line 33-43.

The discussion also has been re-written as suggested and highlighted from line 319-324

3.       

please avoid unssesecary spaces and separations between sentences, but add them where needed ex. Line 348 “….8.80).In the study….”        

Spacing & separations updated.  

4.       

please explain the HRQoL abbreviation in the abstract

Abbreviation explained in line 15-16.

5.       

please shorten the Abstract up to 200 words (please see the journal guidelines) and make it a single paragraph. Moreover, please avoid too many numbers in the abstract and instead focus on words.

The abstract has been re-written and shortened to 203 words focused on the words as commented.

6.       

please avoid subparagraphs/ headliners  in the discussion part

subparagraphs/ headliners  has been deleted

7.       

In Table 2 in Variable Current knee Oa treatment total number of patients is 291 instead of 290. Please double-check the values.

The total number for variable “current knee oa treatment” total checked, it is 290 as shown below:

8.       

In table 4 please change the “Std deviation” for SD. Is it possible to add the intervals +/-?. What is the goal of showing 25-50-75-percentile if you are not commenting/discussing it?

Table 4 has been updated with the minimum and maximum intervals.

Since the 25-50-75 percentile was not discussed, it has been deleted from the table.

9.       

please edit the reference style according to the journal guidelines

All references have been edited as the journal guidelines and adjusted the numbering accordingly.

10.   

please specify reference 4, and if its an online source, the date of access

Reference which was online sources has been updated the referencing

11.   

please edit the citation style in line 94

Reference added as reference no.29 (line 88)

12.   

please change & for and in line 384

Changed done in line 347

13.   

please add references 43,44 and 45

Added the missed 3 references as reference no.40, 41, 42 (line 483-487)

Reviewer 2 Report

Dear authors,

Thank you for the opportunity to review the paper titled, “Does less pain predict better quality of life in mild-moderate knee osteoarthritis patients?”. The purpose of this study was to examine the relationship between the knee functional status and health-related quality of life. The authors provide measures of knee injury and pain and quality of life outcomes; however, the abstract and introduction of the manuscript was difficult to follow. The methods and results were clear but could be improved. The discussion should address the importance of the findings than restating the results.

Title

I recommend changing the title. Using “predict” implies a longitudinal effect, however your study is a cross-section study. Please revise using “associated with” or other words that accurately reflects the findings.

Abstract

Overall, too many acronyms are used, which is extremely difficult to follow.

Clinics and Practices should not have subheadings in the abstract. Please remove.

Line 16:  It will be better to write as “knee OA” than “KOA” as you did throughout your manuscript.

Line 17: HRQoL was not introduced in the abstract but used as an acronym. So, it was difficult to understand why your purpose is to see the relationship between KFS and HRQoL. Please provide a full name.

Line 22: KOOS and SF-36 “were” used? Please check the grammar.

Line 32: Since this is a cross-sectional study, the cause and effect could be not tested. Thus, better HRQoL can be achieved by pain control is not scientifically accurate. Please refine the sentence.

Line 33: Same here. The relationship between MCS and KFS does not “improve” HRQoL. Please refine the sentence.

Introduction

I think relevant information is in the introduction, but disorganized. I suggest you revise the paragraph with topic sentences.

Line 40: So, does it mean OA is the first cause of disability caused by musculoskeletal disorders? Difficult to follow.

Line 42: “Of these”? What of these? Meaning all OA?

Line 50: Does not flow well in the sentence.

Line 51: Since globesity is more informally used and not the main topic of this paper, I suggest replacing with “obesity”.

Line 51-52: fits better in the first paragraph. The second paragraph should focus on knee OA association with quality of life.

Line 74-80: From reading this paragraph, the purpose of this study seems to be measuring the relationship specifically in Malyasian population. I suggest adding “Malaysian” in the title for accuracy.

Materials and Methods

Line 171: What was the primary question for using previous publication for calculating the current sample size?

Discussion

Unnecessarily divided the paragraphs throughout.

Using too many acronyms does not help readers.

The results were redundantly described throughout the discussion. Please discuss your results with the significance of the findings.

Author Response

Dear Reviewer 2, thank you for your comments. Attached are the responses. Thank you. 

Response to Reviewer 2

Comment

Response

Abstract

Overall, too many acronyms are used, which is extremely difficult to follow.

The acronyms used have been minimised

Clinics and Practices should not have subheadings in the abstract. Please remove.

Subheadings in abstract have been removed

Line 16:  It will be better to write as “knee OA” than “KOA” as you did throughout your manuscript.

KOA changed to knee OA as suggested

Line 17: HRQoL was not introduced in the abstract but used as an acronym. So, it was difficult to understand why your purpose is to see the relationship between KFS and HRQoL. Please provide a full name.

Full name of HRQoL has been provided

Line 22: KOOS and SF-36 “were” used? Please check the grammar.

Grammar has been corrected in line 20.

Line 32: Since this is a cross-sectional study, the cause and effect could be not tested. Thus, better HRQoL can be achieved by pain control is not scientifically accurate. Please refine the sentence.

This sentence has been refined.

Line 33: Same here. The relationship between MCS and KFS does not “improve” HRQoL. Please refine the sentence.

This sentence has been refined.

Introduction

I think relevant information is in the introduction, but disorganized. I suggest you revise the paragraph with topic sentences.

The introduction has been simplified and

organized

Line 40: So, does it mean OA is the first cause of disability caused by musculoskeletal disorders? Difficult to follow.

The first sentence that says ‘OA is a leading cause of disability in the world’ has been removed and rearrange to make it clear (line 40-43)

Line 42: “Of these”? What of these? Meaning all OA?

This line has been changed to ‘Among various types of OA, 83% is represented by knee OA’ (line 35)

Line 50: Does not flow well in the sentence.

Line 51: Since globesity is more informally used and not the main topic of this paper, I suggest replacing with “obesity”.

The sentence has been changed to ‘With increasing survival age of the world population and increasing obesity, knee OA is the major reason for knee replacement.’(line 42-43)

The word globesity has been replaced with obesity.(line 43)

Line 51-52: fits better in the first paragraph. The second paragraph should focus on knee OA association with quality of life.

This sentence has been moved to the first paragraph as suggested. Thank you.

Line 74-80: From reading this paragraph, the purpose of this study seems to be measuring the relationship specifically in Malyasian population. I suggest adding “Malaysian” in the title for accuracy.

Title changed to “Does less pain predict better quality of life among Malaysian with mild-moderate knee osteoarthritis?”

Materials and Methods

Line 171: What was the primary question for using previous publication for calculating the current sample size?

The primary question for using the previous publication was ‘the health-related quality of life of knee osteoarthritis patients in a cohort patients attending primary care clinics in Malaysia as most of those attending these clinics have mild to moderate knee OA’

Discussion

Unnecessarily divided the paragraphs throughout.

Headings were removed

Using too many acronyms does not help readers.

Acronyms changed to full terms. E.g physical component summary or mental component summary

The results were redundantly described throughout the discussion. Please discuss your results with the significance of the findings.

Redundant results in discussion have been removed (as highlighted)
